# A Case Study of Polyetheretherketone (II): Playing with Oxygen Concentration and Modeling Thermal Decomposition of a High-Performance Material

**DOI:** 10.3390/polym12071577

**Published:** 2020-07-16

**Authors:** Aditya Ramgobin, Gaëlle Fontaine, Serge Bourbigot

**Affiliations:** CNRS, INRAE, Centrale Lille, UMR 8207—UMET—Unité Matériaux et Transformations, University Lille, F-59000 Lille, France; aditya.ramgobin@univ-lille.fr (A.R.); gaelle.fontaine@univ-lille.fr (G.F.)

**Keywords:** kinetic analysis, polymer decomposition, high performance polymers, simulation

## Abstract

Kinetic decomposition models for the thermal decomposition of a high-performance polymeric material (polyetheretherketone, PEEK) were determined from specific techniques. Experimental data from thermogravimetric analysis (TGA) and previously elucidated decomposition mechanisms were combined with a numerical simulating tool to establish a comprehensive kinetic model for the decomposition of PEEK under three atmospheres: nitrogen, 2% oxygen, and synthetic air. Multistepped kinetic models with subsequent and competitive reactions were established by taking into consideration the different types of reactions that may occur during the thermal decomposition of the material (chain scission, thermo-oxidation, char formation). The decomposition products and decomposition mechanism of PEEK which were established in our previous report allowed for the elucidation of the kinetic decomposition models. A three-stepped kinetic thermal decomposition pathway was a good fit to model the thermal decomposition of PEEK under nitrogen. The kinetic model involved an autocatalytic type of reaction followed by competitive and successive nth order reactions. Such types of models were set up for the evaluation of the kinetics of the thermal decomposition of PEEK under 2% oxygen and in air, leading to models with satisfactory fidelity.

## 1. Introduction

Polymeric materials possess outstanding mechanical and chemical properties, which is why they are being increasingly used in many industries such as aerospace and transport. However, most of them possess a major drawback: heightened flammability. Indeed, the thermal decomposition of organic polymeric materials usually releases small hydrocarbons which can fuel fires. It is therefore imperative to either flame retard them or use inherently fire-retardant polymers.

One such polymer that exhibits such an intrinsic flame-retardant behavior is polyetheretherketone. The thermal decomposition and fire behavior of this material was studied in the first part of this work [1] whereby the decomposition mechanisms were used to explain the enhanced fire properties of the material.

In order to further this study, it was deemed worthwhile to understand the key kinetic parameters that govern the thermal decomposition of these materials. Indeed, it was observed that the presence of oxygen may have very different effects on the onset of the thermal decomposition of a material. Therefore, investigating the effect that different oxygen concentrations may have on the kinetic parameters of the thermal decomposition of these materials can allow the prediction of their behavior in a flaming or nonflaming combustion. Furthermore, the conditions at the polymer surface are either completely pyrolytic, oxygen deprived, or atmospheric.

In our previous work, we have evidenced that the thermal behavior of PEEK is very different depending on the oxygen concentration [1]. There are several steps involved in the thermal decomposition of PEEK, and the steps differ greatly whether the thermal stress is oxygen-rich or deprived. Therefore, in order to complement the thermal decomposition of PEEK in a fire scenario, kinetic models corresponding to the decomposition in oxygen-free, oxygen-poor, and oxygen-rich (air) atmosphere need to be investigated.

The thermogravimetric (TG) plots performed on PEEK at different oxygen concentrations (Figure 1) show that the first mass loss of PEEK is apparently independent of oxygen concentration whereby, the subsequent mass-loss steps are highly dependent on the oxygen concentration. Indeed, it can be seen on the TG plots, at higher temperatures (above 600 °C), the mass loss increases with increasing oxygen concentration. This suggests that the thermal decomposition mechanism is also highly dependent on the oxygen concentration, even if it is as low as 2%.

Therefore, this paper deals with the investigation of the kinetic parameters for the thermal decomposition of PEEK. Three different oxygen levels have been studied: 0%, 2%, and 20%. Kinetic models were devised in order to match the decomposition behaviors of the materials under the previously mentioned atmospheres. Furthermore, the thermal decomposition kinetics of PEEK have been previously studied using dynamic TGA in air or under nitrogen and using an integral model-free analyses such as the Ozawa−Flynn−Wall method [2,3] or using the Coats−Redfern models [4,5]. While these types of analyses provide valuable information concerning kinetic parameters of the material, they assume that the thermal decomposition is a single step reaction, with one activation energy. In order to integrally model the thermal decomposition of a material, it is imperative that a model taking into account the different steps of the thermal decomposition be used.

## 2. Materials and Methods

### 2.1. Thermogravimetric *Analysis* (TGA)

Thermogravimetric analyses (TGA) were conducted on a Netzsch Libra instrument. Powdered samples of 9–10 mg (according to the good practice of TGA, it can be assumed the samples are thermally thin) were placed in open alumina pans and heated up to 900 °C under different percentages of oxygen and nitrogen at different heating rates (1, 2, 5, and 10 °C/min) up to 900 °C. The total gas (nitrogen + oxygen) flow rate was 100 mL/min.

### 2.2. Kinetic Analysis

Kinetic analysis and modeling of the degradation of the samples were made using a Kinetics Neo software package developed by Netzsch Company. The principle has been discussed by Opfermann in [6] and here we only briefly remind the reader of the basic concepts of the method.

For kinetic analysis, it is assumed that the material decomposes according to Equation (1):(1)$$\begin{array}{c} A_{\mathrm{solid}}\to B_{\mathrm{solid}\;\mathrm{or}\;\mathrm{liquid}}+C_{\mathrm{gas}} \end{array}$$

The rate expression de/dt, where e is the concentration of educt (reactant), is assumed to be defined by Equation (2):(2)$$\begin{array}{c} \frac{\mathrm{de}}{\mathrm{dt}}=k\left(T\right)\times f\left(e,p\right)\;\; \end{array}$$
where k is the kinetic constant, p is the concentration of the product, *k* = *A*exp(−*E*/R*T*) according to the Arrhenius law, A is the frequency factor, E is the activation energy and f(e,p) is the so-called ‘‘reaction equation’’ or in the case of TGA, the ‘‘reaction model”.

All reactions are assumed to be irreversible. In the case of degradation and since the evolved gases are continuously removed by the fluid flow in the TGA chamber, this is a reasonable assumption. It is also assumed that the overall reaction (Equation (1)) is the sum of individual reaction steps (formal or true steps) with constant activation energy, as generally accepted in chemistry. The model can then include competitive, independent, and successive reactions. The equations are solved with multivariate kinetic analysis (determination of the parameter via a hybrid normalized Gauss-Newton method or Marquardt method) [7].

By optimizing the models used for the kinetic analysis, the kinetic parameters of each step in the thermal decomposition model of the material can be extracted, allowing for a better understanding of the thermal decomposition behavior of the material.

The approach used to model the decomposition behavior of PEEK was similar to that adopted by Moukhina et al. A model-free analysis was used to determine the initial kinetic decomposition parameters and gain insight regarding the number of steps involved and the types of kinetic models that govern the decomposition [8]. To this approach, we added our own understanding of the thermal decomposition based on the previous work, in which the thermal decomposition mechanism was attempted [1].

The reflection behind the methodology that we have adopted is briefly summarized hereunder.

In order to model the kinetic degradation of a polymeric material, two separate functions can be assumed. One being temperature dependent (*K*(*T*)), and the other governed by the conversion, *α*, *f*(*α*). The latter can be of any value from 0 (no degradation) to 1 (complete degradation). Therefore, the differential equation that defines the kinetics of thermal degradation can be written as Equation (3) [9].
(3)$$\begin{array}{c} \frac{d\alpha }{dt}=K\left(T\right)f\left(\alpha \right) \end{array}$$

$\frac{d\alpha }{dt}$ is the rate of degradation, *K*(*T*) is the temperature dependent rate constant, and f(α) corresponds to the reaction model. It is accepted that K(T) can be described by the Arrhenius equation (Equation (4)):(4)$$\begin{array}{c} K\left(T\right)=Ae^{-\left(\frac{E}{RT}\right)} \end{array}$$
where R is the universal gas constant, E, the activation energy, and A, the pre-exponential factor [10].

The time dependence of Equation (4) can be eliminated by using a constant heating rate $\beta =\frac{d\alpha }{dt}$, by dividing by it (Equation (5)).
(5)$$\begin{array}{c} \frac{d\alpha }{dT}=\frac{A}{\beta }f\left(\alpha \right)e^{-\frac{E}{RT}}\; \end{array}$$

Linearizing Equation (3) leads to obtainable kinetic parameters (*A* and *E*) by using the Equation (6).
(6)$$\begin{array}{c} \ln\left(\frac{\frac{d\alpha }{dt}}{f\left(\alpha \right)}\right)=\ln\left(\frac{A}{\beta }\right)-\left(\frac{E}{RT}\right) \end{array}$$

One approach for kinetic modelling involved the assumption that the activation energy and the pre-exponential factor is constant. A well-known technique that uses this method is the Friedman method, whereby the activation energy and pre-exponential factor are obtained by plotting the logarithmic form of the rate equation of each heating rate (Equation (7)). *α* represents the value at a certain degree of conversion, and *i* the data from the corresponding heating rate experiment [11].
(7)$$\begin{array}{c} \ln\left[\beta_{i}{\left(\frac{d\alpha }{dT}\right)}_{\alpha ,i}\right]=\ln\left(A_{\alpha }f\left(\alpha \right)\right)-\frac{E_{\alpha }}{RT_{\alpha ,i}}) \end{array}$$

The activation energy at particular conversion degrees can be calculated with linear regression from a plot of $\ln\left[\beta_{i}{\left(\frac{d\alpha }{dT}\right)}_{\alpha ,i}\right]$ against $\frac{1}{T_{\alpha ,i}}$ for the heating rates that were used. The plot can provide confirmation as to whether there is more than one step involved in the degradation process. Moreover, the nature of the decomposition step can also be intuitively guessed by comparing the slope of the constant heating rate data [12]. By comparing the magnitudes at the peak slope (the one that is on the right side of the peak) and that of isoconversion lines, three types of reactions are defined: normal, accelerated, and retarded (Figure 2).

A normal reaction corresponds to the curve whereby the magnitude of the peak slope (slope to the right of the peak) and that of the isoconversion lines are of the same magnitude. An accelerated reaction is one where the peak slope is steeper than that of the isoconversion lines. Contrarily, a retarded reaction has a peak slope which is gentler than that of the isoconversion lines. However, one of the major limitations of this method is that it does not cater for the possibility of competitive parallel reactions that may occur during the thermal decomposition process.

It should be noted that for the reaction model to make physical sense, reaction orders above three are not considered. However, because of the complexity of the reactions occurring during the thermal decomposition of a polymeric material, optimizations based on experimental data can lead to noninteger values of reaction orders. This often happens when a step involves more than one pathway towards the same decomposition product.

Another similar method that is used for model-free analyses of kinetic degradation is the Ozawa−Flynn−Wall integral isoconversional method [2,13].

The Ozawa−Flynn−Wall analysis involves an integral method for the calculation of the kinetic parameters, therefore, there is no separation of variables involved. As a result, competitive reactions show variations in activation energies between the Ozawa−Flynn−Wall and the Friedman analyses [12]. This will be helpful in determining the nature of the steps involved in the thermal decomposition of the materials investigated.

Moreover, the insight regarding the thermal decomposition behavior of the polymeric materials will be used in order to devise an experiment-based model, aided by the model-free analysis detailed above.

There are several reaction types that can be attributed to a decomposition step. The typical homogenous reactions and classic solid reactions are listed in Table 1.

The models were optimized using the KineticsNeo software (Netzsch). We have attempted to make kinetic models with the lowest number of steps that gave an acceptable fit and that were consistent with the thermal decomposition mechanism of the material studied.

## 3. Results and Discussion

In the context of this work, model-based analysis for the thermal decomposition kinetics of PEEK was carried out, using dynamic TGA at four different heating rates (1 K/min, 2 K/min, 5 K/min, and 10 K/min) under the three aforementioned atmospheres (nitrogen, 2% oxygen and synthetic air). This allowed for the simulation of the thermal behavior of PEEK under different heating rates. It also provided the kinetic parameters and contribution of each of the thermal decomposition steps that were occurring during the thermal breakdown of the material.

### 3.1. PEEK Decomposition under Nitrogen

TG curves and their corresponding derivative (DTG) curves of PEEK for the four different heating rates of PEEK under nitrogen are shown in Figure 3.

As a high-performance polymeric material, PEEK is highly stable at intermediate temperatures (up to around 450 °C). The temperature at the onset of the decomposition increased with increasing heating rate and involved a sharp mass-loss step, which was followed by another smaller mass loss of moderate slope (slow mass-loss rate). The two steps can be seen on the DTG curve (Figure 3). From it, a sharp peak was observed, corresponding to the first mass loss. The second mass-loss process was deduced by the non-zero derivative value on the DTG curve. This suggests that there are at least two steps involved in the thermal decomposition of PEEK.

From the DTG curves there seemed to be a trend whereby the peak mass-loss rate decreased with increasing heating rate from 1 K/min to 5 K/min. However, this was not the case for the heating rate at 10 K/min, suggesting that there might be a different mechanism at higher heating rates. Therefore, in order to have a better insight concerning the activation energies regarding the thermal decomposition under nitrogen, a model-free analysis of the TG curves was performed, and the Friedman plot is shown on Figure 4.

From the Friedman analysis under nitrogen (Figure 4), it could be observed that the reaction is multistepped. Two peaks were clearly discernible from the heating rates at 1, 2, and 5 K/min. However, at 10 K/min, there was less obvious discernibility of the second peak, suggesting that the thermal decomposition mechanism might be different at higher heating rates. From the isoconversion lines (shades of red to blue lines corresponding to a particular conversion degree), it was clear that the peak slope was much steeper than the isoconversion lines, suggesting an accelerated reaction. An autocatalytic decomposition model might explain this first step of decomposition.

Moreover, from the Friedman analysis, the activation energy and the pre-exponential factor with respect to the conversion can be calculated (Figure 5).

It could be observed that the activation energy was not constant throughout the decomposition process. This, coupled with the DTG curves, suggests that there are at least three steps in the thermal decomposition of PEEK under nitrogen. While two peaks were seen on the DTG curves, the activation energy plot showed that there were at least three steps—the first at the beginning of the conversion (0–60%), the second whereby the activation remained relatively constant until around 60% conversion, and a third, corresponding to the peak seen at around 80% of conversion.

It should be noted that a 100% conversion means that the whole decomposition has been achieved. This hypothesis was not always verified in the case of our investigation. However, seeing how the DTG was almost constant at 800 °C, it could be assumed that the conversion was nearly 100% at that temperature for the corresponding step of the decomposition.

While the DTG plots provided insight into the major steps involved in the thermal decomposition of PEEK, and the Friedman plot allowed the identification of the initial kinetic parameters, they did not take into consideration the possibility of competitive reactions that might occur during the thermal decomposition. To do so, the activation energies calculated using the Friedman method were compared to those obtained using the Ozawa−Flynn−Wall method. The Figure 6**.** shows the activation energy plots from the two methods.

From the two activation energy plots, it could be seen that the first step corresponded to an activation energy around 200 kJ/mol for both plots from a conversion of 0 to around 50%. However, as from a conversion of around 50%, a difference in activation energy was observed. This suggested that there were possibly competitive reactions occurring.

From the insight gathered by the model-free analyses, it was possible to define a model for the decomposition of PEEK under nitrogen. The decomposition used for modeling the thermal decomposition kinetics of PEEK under nitrogen is shown on Scheme 1. The explanations of each step are detailed in the subsequent paragraphs.

From the mechanism of the thermal decomposition of PEEK under nitrogen, it was seen that the first reactions were the random scission of the polymeric chain. Therefore, for our kinetic model, random scission, leading to char formation, should correspond to the first step of the reaction (A–B on Scheme 1). Moreover, from the Friedman plot, this step was assigned to an accelerated reaction. Therefore, an autocatalyzed reaction model was be adopted for this step. This can be explained by the fact that random scission of polymeric materials at high temperature leads to the production of highly reactive radical species. These species can in turn react with the remaining polymer to further the decomposition process. In some ways, the product of the reaction is aiding in the first step of the reaction. Therefore, the first decomposition step was assigned to an nth order reaction (where n needs to be optimized) involving an autocatalysis by the product (reaction order to be optimized).

Furthermore, from the Ozawa−Flynn−Wall (OFW) and Friedman plots, it was seen that the second degradation step potentially involved competitive reactions. These reactions involved the rearrangement of the char and further scission reactions, as reported in our previous work [1]. From the DTG curves, it could be seen that the mass-loss rate was much diminished after the first step of the decomposition. Therefore, it was less likely that the decomposition reactions occurring during this step led to the generation of any reactive species that would increase the rate of the reaction. Thus, the competitive reactions involved in the second step of the thermal decomposition were both modelled by a simpler nth order reaction.

The last step modelled was assigned to the decomposition of the char. It corresponded to very little mass loss and had a very high activation energy (Figure 6) as it involved the decomposition of a very stable structure.

A summary of the optimized parameters based on the kinetic model above are shown on Table 2.

The first step of the thermal decomposition of PEEK under nitrogen was assigned to an autocatalytic reaction, with an activation energy of around 208 kJ/mol, a reaction order of 1.5 and an autocatalytic order of 1.5. This activation energy for the first step was in accordance with the Friedman analysis whereby the initial activation energy was calculated to be around 220 kJ/mol. Subsequently, two competitive decomposition reactions were modelled, the first, with an activation energy of around 226 kJ/mol and a reaction order of 1.2. The other competitive reaction was of the same nature, but a lower order (0.8). It also had a lower contribution (0.085 as compared to 0.556). The final reaction was also an nth order reaction with a reaction order of 2.3. The high activation energy was coherent with the Friedman activation energy plot (Figure 5).

In order to visualize our model, modeled TG curves based on the kinetic parameters in Table 3**.** were plotted on the same axes as the experimental TG curves. (Figure 7).

From a statistical point of view, the experimental curve and the simulated curve had a correlation coefficient of 0.99990. From Figure 7, we saw that the simulated curves were consistent with the experimental ones. Each of the four reactions in the kinetic decomposition model had a contribution above 8%, meaning that they all played a significant role in the thermal decomposition of PEEK under nitrogen. Having modeled the kinetics of the thermal decomposition of PEEK under nitrogen, the method would now be applied to the thermal decomposition of PEEK under 2% oxygen and air.

### 3.2. PEEK Decomposition under 2% Oxygen

From the TG curves at varying oxygen concentrations (Figure 1) it can be inferred that the first decomposition step is only slightly dependent on the onset of the thermal decomposition temperature of PEEK. However, from the temperature at the onset of the decomposition at a heating rate of 10 °C/min, there was no further evidence to support this hypothesis. Therefore, extracting the kinetic parameters concerning the thermal decomposition of PEEK under a slightly more oxidative atmosphere (2% oxygen) would challenge the hypothesis and could either support or contradict it. To do so, the same method as before was used.

The TGA of PEEK at the same heating rates as before was performed under 2% oxygen. The resulting TG curves and their corresponding DTG curves are shown on Figure 8.

Similar to the measurements under nitrogen, and as expected from the previous study on the thermal stability of PEEK [1], its TG curves under a low level (2%) of oxygen also depicted the high stability of PEEK. As expected, the onset of the thermal decomposition temperature increased as the heating rate increased. The onset of decomposition temperatures under low heating rates was lower under 2% oxygen than under nitrogen. In addition, the mass-loss rate for the first decomposition step was much lower at 1 K/min (0.5%/°C) as compared to the other heating rates (>0.8%/°C). On the other hand, the mass-loss rate of the second decomposition step was higher at low heating rates than at high ones, as could be observed on the DTG curve (Figure 8, right). This suggested that the first step of the decomposition might involve a competitive reaction that occurred faster at high temperatures than lower ones.

The first decomposition step was followed by a second one, whereby the mass-loss rate was higher at lower heating rates (except at 1 K/min). This reaction can be assigned to at least two competitive reactions: the oxidation of the char coupled with the subsequent pyrolytic decomposition of the initial char that was formed during the first stage of the decomposition. The higher mass-loss rate at lower heating rates can be explained by the limited oxygen availability, suggesting that thermal oxidation was a limiting factor at high heating rates.

To dig deeper into the kinetics of the thermal decomposition of PEEK under 2% of oxygen, the Friedman analysis corresponding to the aforementioned heating rates was plotted (Figure 9).

From the Friedman analysis (Figure 9) deeper insight concerning the multistepped thermal decomposition of PEEK under 2% oxygen was perceived. Indeed, at high heating rates, it could be observed that the first decomposition reaction corresponded to an accelerated one. This was observed on the slope of the isoconversion lines which was gentler than the first peak slope on the Friedman analyses curves. Two peaks were visible on the Friedman plots at heating rates corresponding to 1 K/min and 2 K/min. The absence of a clear second peak for 5 K/min and 10 K/min on the Friedman plot could be explained by the incompletion of the decomposition at 800 °C.

To further our insight concerning the thermal decomposition of PEEK under 2% oxygen, the activation energy and corresponding pre-exponential factor were plotted, based on the Friedman analysis calculations. The resulting plots are shown on Figure 10.

The activation energy increased in the first part of the conversion and a small shouldering was observed before the subsequent peak at around 800 kJ/mol. This suggested that there were two steps occurring right at the beginning of the decomposition. Moreover, the activation energy for the subsequent decomposition decreased sharply and was even negative. This meant that the mass-loss rate decreased as the temperature was increased. Indeed, on Figure 2, we can observe that the second decomposition step exhibited a higher mass-loss rate at low heating rates as compared to higher heating rates. This suggested that the second decomposition step had a negative dependence on the temperature. From a phenomenological point of view, this could be interpreted as a barrierless reaction step, or as a decomposition reaction that occurs spontaneously. A possible explanation for this barrierless decomposition step could come from the initial decomposition of PEEK. Indeed, it was observed in our previous study that the first decomposition step of PEEK was highly exothermic [1]. The heat released during the initial decomposition of the polymer was not taken into account in the equations used to calculate the activation energies. This heat released was enough to start the second step of the decomposition, which was assigned to a thermo-oxidative decomposition step of PEEK. Furthermore, at conversions above 50%, there seemed to be a relatively constant activation energy for the decomposition, suggesting that a final step was involved.

Valuable insight concerning the minimum number of steps involved in the thermal decomposition of PEEK under 2% oxygen was brought thanks to the model-free analyses. In order to have a better understanding on the possibility of competitive reactions that could be involved during the thermal decomposition of PEEK, the activation energy plot from the Friedman method was compared with that from the OFW method (Figure 11).

By comparing the activation energies from the two model-free methods (Figure 11), it is evident that from 0% conversion to around 50% conversion, the activation energy with respect to the conversion were not the same. This implied that there were competitive reactions occurring during the thermal decomposition of PEEK under 2% oxygen.

From the model-free analyses, we have seen that the decomposition of PEEK adopted a complex pathway involving both competitive and consecutive reactions. In order to model this, a kinetic pathway was required. Moreover, the thermal decomposition pathway that was elucidated in our previous work [1] was required so that a deep understanding of the kinetics of the thermal decomposition of PEEK could be achieved.

A model for the kinetics of the decomposition of PEEK under 2% oxygen is suggested and illustrated in Scheme 2. The details of the model are explained hereafter.

From the thermal decomposition reactions arising during the onset of the decomposition, the first step of the thermal decomposition was one leading to the formation of carbon monoxide, radical-ended polymer chains as well as a charred graphite-like structure. This reaction was assigned as the first step in the thermal decomposition of PEEK under 2% oxygen. Indeed, it was reported that an increase in oxygen partial pressure (therefore concentration) amplified the cross-linking phenomenon in PEEK at temperatures between 380–440 °C [14]. Moreover, the radical ended polymeric chains could react with the undecomposed polymer to further the decomposition reaction. Therefore, from a reaction kinetics perspective, this step was assigned to an autocatalytic decomposition reaction (step 1.1).

Once the charred structure formed, the presence of oxygen meant that it could be thermo-oxidized. However, since the availability of oxygen was relatively low, the governing mechanism that limited the rate of the reaction was the diffusion of oxygen towards the char. Therefore, the kinetic step following the char formation was assigned to a three-dimensional Jander’s type diffusion type (Equation (7)) [15]. Additionally, further decomposition of the char occurred at higher temperatures, along with thermo-oxidation of the remaining polymeric materials. This decomposition step was assigned as an nth order Arrhenius decomposition model.

Moreover, along with the formation of the charred structure, random scission all over the material was also suggested. These scissions led to reactive radicals as well as small molecules. The reactive radicals could further react with the remaining polymer chain to further the thermal decomposition of polymeric materials. This behavior was reminiscent of an autocatalytic reaction as the products were taking part in the initial reaction. Therefore, an autocatalytic reaction was assigned as a competitive reaction to the above-mentioned decomposition (step 1.2). During this step, the small molecules were thermo-oxidized by the oxygen present in the atmosphere. However, it had little impact on the kinetics of the decomposition of PEEK.

Based on the model decomposition pathway in Scheme 2, the kinetic parameters for the thermal decomposition of PEEK under 2% oxygen were optimized. A summary of the kinetic parameters for each step is given in Table 3.

In order to visualize our model, simulated TG curves based on the kinetic parameters in Table 3**.** were plotted on the same axes as the experimental TG curve (Figure 12).

From a statistical point of view, the correlation coefficient between the experimental and the modeled curves was 0.99940.

The multistepped decomposition kinetic pathway was consistent with the thermal decomposition mechanism of PEEK. However, while the presence of oxygen appeared to have only a little influence on the thermal stability of PEEK during the previous investigation [1], the kinetic analysis has shown that the decomposition pathway was highly affected. The main evidence for this was the competitive reaction in the first step of the thermal decomposition of PEEK under 2% oxygen. This step was assigned to a single decomposition step in the kinetic model for the decomposition pathway of PEEK under nitrogen.

It was interesting to note that the experimental TG curve at a heating rate of 10 K/min showed a slightly higher deviation from the simulated one than the other ones. One explanation for this could be that there was a change in the thermal decomposition mechanism at higher heating rates.

Moreover, the thermal decomposition kinetics of PEEK under 2% oxygen have shown that there can be a major change in the kinetic parameters even in the presence of low oxygen concentrations. Therefore, to go further in the investigation of the thermal stability of PEEK, the kinetics of its thermal decomposition in a more aggressively thermo-oxidative atmosphere (air), have been studied. The approach used unfolds in the following section.

### 3.3. PEEK Decomposition in Air

Model-free kinetic analysis of the decomposition behavior of PEEK in air has been reported by Vasconcelos et al. [5]. However, model-free analyses assume that the decomposition is a single step process. It is clear from the TG and DTG curves of PEEK at different heating rates that the decomposition is a multi-stepped one (Figure 13).

Indeed, at a heating rate of 10 °C/min, a first decomposition step was visible on the TG and DTG curve, corresponding to around 30% of mass loss. After this, there was a slight stabilization for a few degrees before another sharp mass loss was recorded. This two-step phenomenon was visible on the TG and DTG curves at high heating rates but became less discernible at low heating rates. However, on the DTG curves, as compared to the TG curves, the multistepped decomposition behavior was clear. A first, minor step at the beginning of the decomposition was observed at 400 °C at 1 °C/min, and at around 500 °C at 10 °C/min. Under heating rates above 2 °C/min, two other major peak mass- loss rates were visible. At 1 °C/min, the heating rate overlapped with the second decomposition step. However, a shouldering in the peak mass-loss rate indicated that the two reactions were occurring in quick succession.

From the DTG and TG curves of PEEK under the different heating rates, we can deduce that the thermal decomposition of PEEK in air occurred with at least three thermal decomposition steps. In order to have a deeper understanding of the nature of the decomposition steps occurring, the Friedman analysis of PEEK based on the TGA above was plotted (Figure 14).

The Friedman analysis (Figure 14) provided information on the different steps that might occur as well as the types of reactions occurring during these steps. The first peak on the right of the graph corresponded to the first decomposition step. When looking at the Friedman plots for heating rates above 1 K/min, two other peaks were visible after this step. This means that there are at least three steps in the kinetic decomposition mechanism of PEEK. The third peak was not clearly visible on the Friedman plot corresponding to 1 K/min because the two peaks were close to each other and were overlapping, making one large peak. To confirm this, the activation energy plot from the same Friedman plot is shown on Figure 15.

The activation energy plot from the Friedman method (Figure 15) clearly showed that the activation energy was not constant throughout the PEEK’s decomposition in air. A first step of the decomposition could be assigned to the beginning onset of the decomposition with an activation energy around 180 kJ/mol (conversion < 10%). After this, another step of decomposition, at about a quarter of the conversion was visible as a shouldering in the activation energy plot and a decrease in the activation energy with respect to the conversion. At 50% of conversion, the activation energy remained relatively constant, suggesting that there was no further decomposition step until the final 10% of the conversion, where a small increase of activation energy was observed.

From the Friedman analysis, we can conclude that there are at least three major steps in the kinetics of the thermal decomposition of PEEK in air. However, to find out if there were any competitive reactions occurring during the thermal decomposition of PEEK in air, the activation energy of PEEK from the Friedman and OFW method were plotted on the same plot (Figure 16).

The activation energy plot obtained from the Friedman method is very different from that obtained by the OFW method. The major difference comes from the shape of the first and second step, suggesting that there are competitive reactions occurring during the thermal decomposition of PEEK in air from the second step onward. This should be taken into consideration when elaborating the kinetic model.

In order to have a proper kinetic model for the thermal decomposition of PEEK in air, it was essential that it was coherent with the decomposition mechanism of PEEK. In our previous work, we studied the thermal decomposition mechanism of PEEK under inert atmosphere [1]. We also observed that the onset of the decomposition was only slightly dependent on the atmosphere it was in. However, the presence of 20% of oxygen seems to have had an effect at low heating rates. Therefore, the model should be adapted so that it fits this difference. The suggested model considering all information is shown on Scheme 3 and is described hereafter.

Contrary to the thermal decomposition of PEEK under nitrogen, the first step of the thermal decomposition in air comprised an additional first step, whereby a small mass loss was recorded. This reaction step had a small contribution to the whole decomposition process but had to be taken into consideration for the kinetic model. This decomposition reaction probably corresponded to the thermo-oxidation of the crosslinked, charred structure formed before any mass loss was recorded [14].

Following this step, a thermal decomposition similar to the thermal decomposition of PEEK under nitrogen was observed on the TG curves. This deduction was made due to the similarities in the shape of the main decomposition step from the TG plots of PEEK in air and under nitrogen (Figure 1). This step was assigned to the formation of the char, which was formed by an autocatalytic reaction phenomenon. This is why an autocatalytic decomposition model was considered for this step. Indeed, the thermal decomposition of PEEK proceeded by random scission of the polymer chain, causing the release of reactive radicals. These radicals could further react with the unreacted polymer chain to further the decomposition process. It is interesting to note that the order of this reaction was optimized to 0.5, and the autocatalysis order was 1.6. This means that species released during the thermal decomposition play a bigger role in the kinetics of the decomposition than the initial product.

However, the comparison between the activation energy plots from the Friedman analysis and the OFW analysis suggested that there was possibly a competitive reaction occurring at this stage. Indeed, the presence of oxygen could cause the thermo-oxidation of the polymer, independently of the “concentration” of the polymer. On the other hand, the products that were formed after the thermo-oxidative decomposition led to the formation of reactive species that could further cause the decomposition of the material. This could also be translated by an autocatalytic effect on this step of the thermal decomposition of PEEK.

The third step of the thermal decomposition also involved competitive reactions (Figure 16). It involved the decomposition of the char that was formed in step two. However, unlike under nitrogen, it is well known that the thermo-oxidative stability of a charred structure is much less than its pyrolytic stability. Therefore, the activation energy for the decomposition of the char was much less in this model as compared to that in the decomposition model of PEEK under nitrogen.

The results of the aforementioned kinetic parameters are detailed in Table 4.

In order to visualize our model, modeled TG curves based on the kinetic parameters in Table 4 were plotted on the same axes as the experimental TG curves. (Figure 17).

From a statistical perspective the correlation between the experimental and the simulated TG curves is 0.99970.

From the decomposition reactions that were used to model the thermal decomposition of PEEK in air, it can be noted that the autocatalytic nature of the first decomposition step was not present. This is explained by a thermo-oxidation reaction whereby the decomposition products do not contribute to further the decomposition. However, the subsequent decomposition reactions both involved an autocatalytic factor, both with non-negligible contributions (>10%). This suggests that there are reactive decomposition products that are formed during these reactions. These reactions are reminiscent of the char formation reactions that were attributed for the kinetic model for the thermal decomposition of PEEK under nitrogen. Finally, the last step of the thermal decomposition consisted of two competitive reactions with a combined contribution of 0.325. This corresponded to the oxidation of the yet undecomposed char that was formed during the initial decomposition steps.

## 4. Conclusions

The kinetic parameters for the thermal decomposition of PEEK under three different oxygen levels have been calculated. We have seen that the kinetic decomposition pathway adopted by PEEK is very different even in early stages of the decomposition. This means that despite the apparent independence of oxygen concentration that was postulated in the first part of this work the presence of oxygen does have a significant effect on the decomposition of PEEK [1].

Thanks to our previous work which aimed at elucidating the different chemical reactions occurring during the pyrolysis of PEEK, we were able to design and optimize a comprehensive kinetic model for the thermal decomposition under nitrogen. Moreover, by using the previously acquired insight concerning the influence of oxygen in the decomposition and fire behavior of PEEK, comprehensive kinetic models for the thermal behavior of PEEK under low oxygen concentrations as well as in air were also elaborated.

Under nitrogen, the onset of the thermal decomposition is assigned to an autocatalytic decomposition mechanism. This is also the case for the onset of the thermal decomposition under 2% oxygen. However, in this case, oxygen brings about a competitive degradation reaction, also assigned to an autocatalytic decomposition reaction. Finally, under air, the first decomposition step involves another type of reaction. A small mass loss right at the onset of the decomposition suggested that the abundant presence of oxygen favored another initial decomposition.

The complexity of the subsequent decomposition reactions varied, depending on the atmosphere in which the polymer was. Indeed, the difference in abundance of oxygen gave way to different reaction pathways, whereby thermo-oxidation occurs. The kinetics of the thermal decomposition of PEEK under different oxygen levels have provided insight into how it may behave under different areas in a fire scenario. A burning material usually undergoes pyrolysis in the absence of oxygen and a material subjected to an incident heat flux in a well-ventilated room under fire undergoes thermo-oxidative decomposition in air. However, in a less ventilated fire scenario, the thermal decomposition of the material occurs in oxygen depletion. Therefore, this investigation allowed for characterization of the kinetics of the thermal decomposition behavior of PEEK in a fire scenario. These kinetic models can be adapted and integrated to fire engineering models in order to have a more elaborate and more realistic model fire scenario than that which is presently used.

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
