# Peer review of "A Case Study of Polyetheretherketone (II): Playing with Oxygen Concentration and Modeling Thermal Decomposition of a High-Performance Material"

_polymers, 2020, doi:10.3390/polym12071577_

Round 1

Reviewer 1 Report

Interesting article, relevant to a specialized audience. Very well written. Logical analysis and discussion. The description of thermal degradation reactions under nitrogen and corresponding proposed mechanisms were very well described. The kinetic aspects of degradation under 2% Oxygen and air were well presented, but detailed chemical reaction mechanisms weren’t proposed. In my opinion, the authors should justify the lack of reaction details in the text (perhaps those are initial steps of a broader study?). In other words, the authors should attempt to propose/suggest/discuss structures for the letters “A”, “B”, “C”, etc in Schemes 2 and 3.

Flow rate for TGA?

Pg 3, line 89 – The authors refer to eq (9) which is not presented in the text.

Figure 2 – what do the colors mean (green, red, and blue) – the info needs to be included in the Figure caption.

Figure 4 – I imagine “red” corresponds to 10K/min instead of 5K/min.

Figure 12 – It appears that a greater deviation between experimental and calculated data occurs for 10K/min, especially at higher temps (>600ËšC). I think the authors should briefly discuss the possibility of a mechanism change as a function of heating rates.

Pg 16, line 422 – According to Figure 13, the first degradation step at 1K/min occurs at 450 ËšC, and not 400 ËšC as mentioned in the text.

Author Response

Reviewer 1:

Interesting article, relevant to a specialized audience. Very well written. Logical analysis and discussion. The description of thermal degradation reactions under nitrogen and corresponding proposed mechanisms were very well described. The kinetic aspects of degradation under 2% Oxygen and air were well presented, but detailed chemical reaction mechanisms weren’t proposed. In my opinion, the authors should justify the lack of reaction details in the text (perhaps those are initial steps of a broader study?). In other words, the authors should attempt to propose/suggest/discuss structures for the letters “A”, “B”, “C”, etc in Schemes 2 and 3.

Thank you for these sensible remarks. In fact, the detailed chemical reactions and the study of the thermal decomposition mechanism of PEEK has attempted in our previous paper, which is still under review in this journal (we are expecting it will be accepted soon).

Flow rate for TGA?

The total flow rate for the TGA was 100 mL/min. This has been added to the experimental section.

Pg 3, line 89 – The authors refer to eq (9) which is not presented in the text.

Thank you for this remark, we meant equation 1. This has been corrected in the text.

Figure 2 – what do the colors mean (green, red, and blue) – the info needs to be included in the Figure caption.

The three colors correspond to three different constant heatings rates. The original publication does not give an actual value because the figures are used to explain the reasoning in a similar way that we have done in this article.

Figure 4 – I imagine “red” corresponds to 10K/min instead of 5K/min.

Thank you for this remark. We have corrected this in the article.

Figure 12 – It appears that a greater deviation between experimental and calculated data occurs for 10K/min, especially at higher temps (>600ËšC). I think the authors should briefly discuss the possibility of a mechanism change as a function of heating rates.

Thank you for this sensible comment. We have added a small paragraph regarding this.

Pg 16, line 422 – According to Figure 13, the first degradation step at 1K/min occurs at 450 ËšC, and not 400 ËšC as mentioned in the text.

Thank you for this comment. However, while the major decomposition step does start at 450 °C, there is a small decomposition step that is visible (more clearly on the DTG) beginning at 400 °C.

Reviewer 2 Report

The research article presented here helps to understand the flammability characteristics of materials. The work portraited here appeals to both the scientific and student community. Subsequently the work can be accepted for communication in the present form.

Author Response

The research article presented here helps to understand the flammability characteristics of materials. The work portraited here appeals to both the scientific and student community. Subsequently the work can be accepted for communication in the present form.

Thank you for these kind remarks.

Reviewer 3 Report

This work mainly introduces the establishment of kinetic model for PEEK decomposition under different atmospheres. This work is well written and very interesting. I believe this work could provide useful suggestions for research and industrial works.

I have only one suggestion that authors have better combine the figures which are with relations. Seventeen figures are too many to easily and comfortably read.

Author Response

This work mainly introduces the establishment of kinetic model for PEEK decomposition under different atmospheres. This work is well written and very interesting. I believe this work could provide useful suggestions for research and industrial works.

I have only one suggestion that authors have better combine the figures which are with relations. Seventeen figures are too many to easily and comfortably read.

While it is definite that the number of figures is quite high, we believe that the sequence of figures and explanation is essential to provide a simple and direct explanation to ease the understanding and readability of the paper. For instance, it is essential to separate the activation energy/Log(A) plot based on Friedman analysis for a first analysis before comparing the activation energy plot based on the Friedmann method with that of the Ozawa-Flynn-Wall method. However, if the reviewer is convinced that it is essential that the figures are combined for a comfortable read, we could do it.